# Intratumoral CD40 activation and checkpoint blockade induces T cell-mediated eradication of melanoma in the brain

Manisha Singh[1], Christina Vianden[1], Mark J. Cantwell[2], Zhimin Dai[1], Zhilan Xiao[1], Meenu Sharma[1], Hiep Khong [1], Ashvin R. Jaiswal[3], Faisal Faak[1], Yared Hailemichael[1], L.M.E. Janssen[1], Uddalak Bharadwaj[4], Michael A. Curran[3], Adi Diab[1], Roland L. Bassett[5], David J. Tweardy[4], Patrick Hwu[1,6] & Willem W. Overwijk[1,3,6]

CD40 agonists bind the CD40 molecule on antigen-presenting cells and activate them to prime tumor-specific CD8[+] T cell responses. Here, we study the antitumor activity and mechanism of action of a nonreplicating adenovirus encoding a chimeric, membrane-bound CD40 ligand (ISF35). Intratumoral administration of ISF35 in subcutaneous B16 melanomas generates tumor-specific, CD8[+] T cells that express PD-1 and suppress tumor growth. Combination therapy of ISF35 with systemic anti-PD-1 generates greater antitumor activity than each respective monotherapy. Triple combination of ISF35, anti-PD-1, and anti-CTLA-4 results in complete eradication of injected and noninjected subcutaneous tumors, as well as melanoma tumors in the brain. Therapeutic efficacy is associated with increases in the systemic level of tumor-specific CD8[+] T cells, and an increased ratio of intratumoral CD8[+] T cells to CD4[+] Tregs. These results provide a proof of concept of systemic antitumor activity after intratumoral CD40 triggering with ISF35 in combination with checkpoint blockade for multifocal cancer, including the brain.

---

[1] Department of Melanoma Medical Oncology, The University of Texas MD Anderson Cancer Center, Houston, TX 77054, USA. [2] Memgen, LLC Houston, TX 77046, USA. [3] Department of Immunology, The University of Texas MD Anderson Cancer Center, Houston, TX 77054, USA. [4] Department of Infectious Diseases, Infection Control and Employee Health, The University of Texas MD Anderson Cancer Center, Houston, TX 77054, USA. [5] Department of Biostatistics, The University of Texas MD Anderson Cancer Center, Houston, TX 77054, USA. [6] The University of Texas MD Anderson Cancer Center, UTHealth Graduate School of Biomedical Sciences at Houston, Houston, TX 77054, USA. Correspondence and requests for materials should be addressed to W.W.O. (email: WOverwijk@mdanderson.org)

Although surgical resection is a reliable treatment for localized melanoma, treatment options for metastatic melanoma are limited. Brain metastasis is a major clinical problem in patients with advanced melanoma[1], and the incidence of brain metastasis is increasing yearly. Immunotherapies, like T cell checkpoint blockade with anti-CTLA-4 and anti-PD-1 antibodies, have improved the median survival of patients with metastatic melanoma and brain melanoma. However, the majority of patients are still not cured by these therapies[2], leaving a need for more effective melanoma therapy. One strategy to enhance the efficacy of checkpoint blockade therapy is to increase the frequency of tumor-specific T cells, for example, by antitumor vaccination.

A critical aspect for successful tumor vaccines is the selection of suitable tumor antigens. However, it can be difficult to find antigens with high, tumor-restricted expression in a sufficiently large fraction of patients to allow for development of a commercially viable vaccine therapy[3]. An alternative strategy is to use immunomodulators that directly activate innate and adaptive immune cells in the tumor microenvironment, facilitating the generation of T cells against often-unique neoantigens encoded by tumor-specific mutations. We and others have reported that intratumoral immunotherapies induce systemic, tumor-specific T cell responses that can target metastases, distant from the initially treated tumor mass, making this a promising approach for the treatment of metastatic cancers[4–6].

Activation of tumor-specific T cell responses has been shown to require activation of the CD40 receptor on antigen-presenting cells[7]. In this regard, agonist CD40 antibody and the cognate CD40 ligand (CD40L) are candidates for tumor immunotherapy. Preclinical and clinical studies with agonist CD40 antibody have shown induction of antitumor immune responses and evidence of efficacy[7, 8]. However, systemically delivered CD40 agonists have resulted in various adverse effects during clinical testing, such as cytokine release syndrome and organ-specific toxicities[9].

ISF35 is a nonreplicating adenovirus encoding a human–mouse chimeric, optimized form of CD40L that is a potent CD40 agonist[10]. ISF35 is delivered by intratumoral injection, resulting in membrane-bound CD40L expression that targets the CD40 receptor on antigen-presenting cells at the site of injection. Cell-surface trimeric CD40L, such as ISF35, results in enhanced CD40 receptor clustering, critical for optimal immune activation. This localized intratumoral ISF35 expression has not caused off-target adverse events such as cytokine release

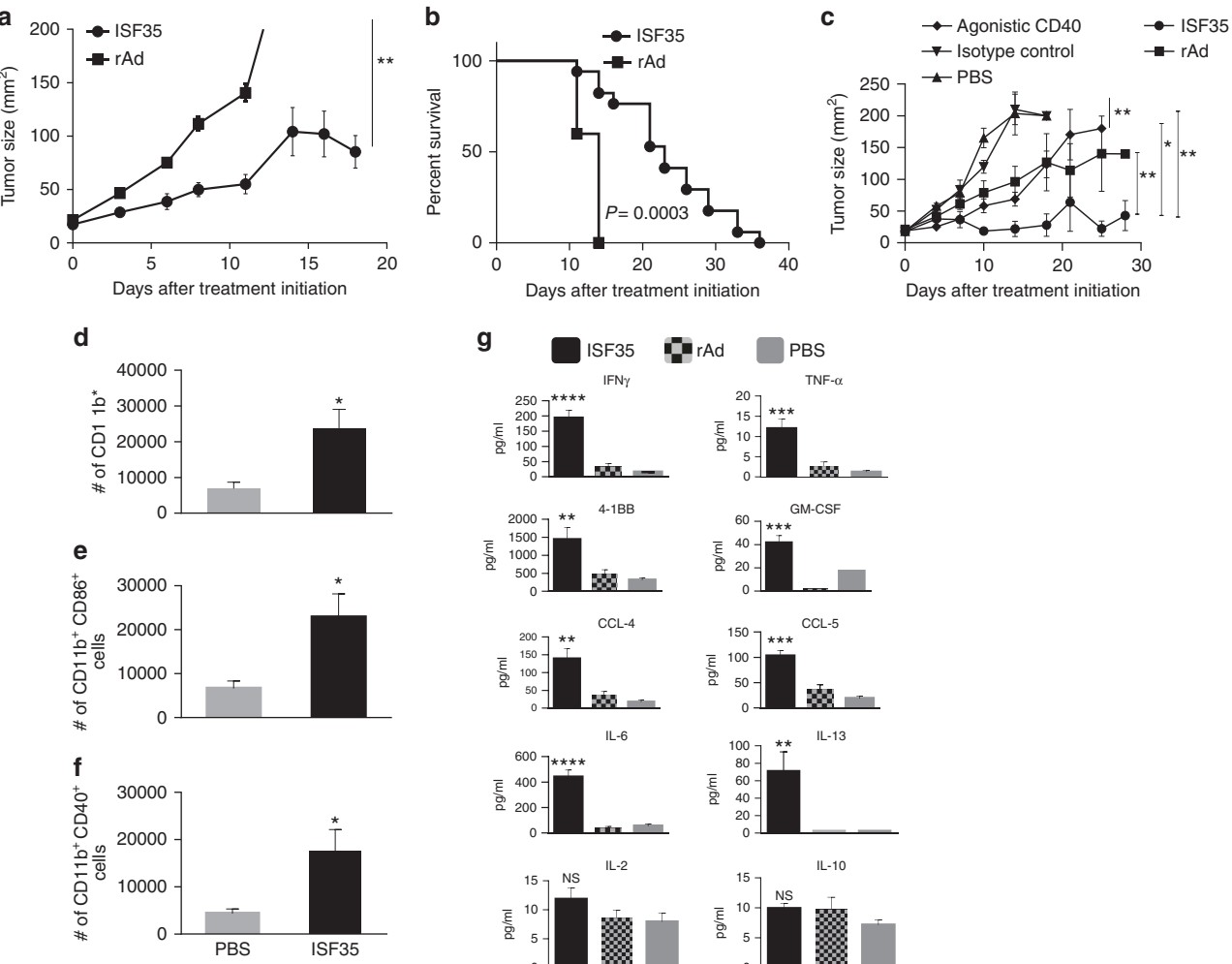

**Fig. 1** Antitumor activity and immune response upon intratumoral ISF35 treatment. Mice bearing 8-day s.c. B16.F10 tumors were treated on day 0, 4, 8, and 12 with ISF35 or control rAd/PBS. **a** Tumor growth ($n = 10$). **b** Mouse survival ($n = 10$). **c** Tumor growth after i.t. treatment with agonistic CD40 mAb or ISF35 ($n = 8$). **d-f** Myeloid cell numbers and activation in tumor 5 days after initiation of ISF35 or PBS treatment. **g** Intratumoral cytokines and chemokines 7 days after initiation of ISF35 or rAd or PBS treatment, measured by Luminex from a tumor supernatant ($n = 5$). Data are representative of at least two independent experiments and analyzed by unpaired two-tailed $t$ test or one-way ANOVA. *$P < 0.05$, **$P < 0.005$, ***$P < 0.0005$ and ****$P < 0.00005$. Error bars are SEM. Survival analysis was performed with the log-rank test

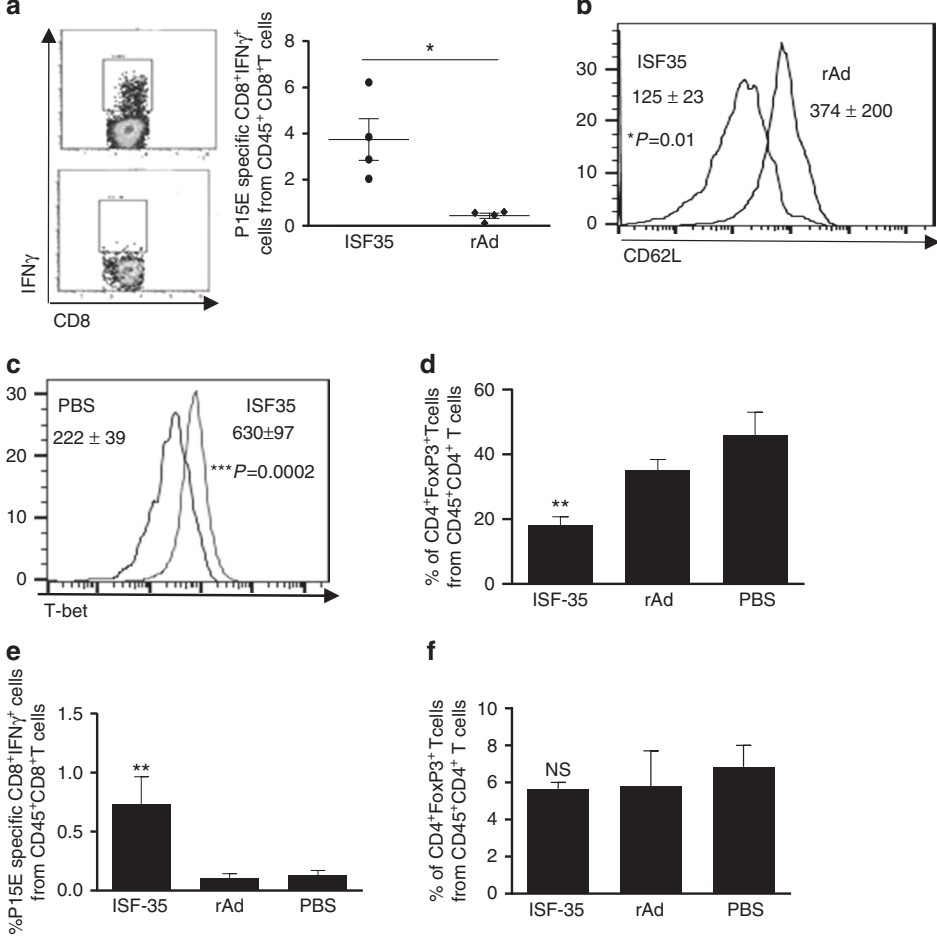

**Fig. 2** Induction of tumor-specific CD8$^+$ T cell immunity and reduction of regulatory CD4 T cells through intratumoral ISF35 treatment. Mice bearing 8-day s.c. B16.F10 tumors were treated as indicated every 4 days and analyzed 9 days after initiation of treatment. **a** Tumor antigen (p15E)-specific IFN-γ-producing CD8 T cells in TIL. Percent of CD8$^+$IFN-γ$^+$ cells (left) and cumulative data (right). **b** CD62L. **c** T-bet expression (MFI) in CD8$^+$ T cells. **d** Percentage of CD4 Treg from CD45$^+$CD4$^+$ T cells in tumor. **e** Tumor-specific CD8 T cells and **f** Tregs in spleen. Data are representative of at least 2 independent experiments and analyzed by unpaired two-tailed $t$ test or one-way ANOVA ($n = 4$ or 5 mice/group). *$P < 0.05$, **$P < 0.005$ and ***$P < 0.0005$. Error bars are SEM

syndrome observed with systemic CD40 mAbs treatment, likely due to its induction of membrane-bound CD40L that is not cleaved into the circulation[11]. ISF35 has previously been tested as a monotherapy and in combination with chemo-immunotherapy for chronic lymphocytic leukemia patients, with signs of clinical efficacy and transient, mild adverse events consisting primarily of flu-like symptoms[12]. This makes intratumoral ISF35 a candidate for the induction of tumor-specific T cells, which could possibly synergize with checkpoint blockade therapy in melanoma and other cancers. To test this hypothesis, we examined the antitumor activity and mechanism of action of intratumoral ISF35 against injected and distant, uninjected tumors. We also examined if intratumoral ISF35 could overcome primary resistance to PD-1 and CTLA-4 checkpoint blockade therapy, including against tumors in the brain.

## Results

**Induction of antitumor immunity after ISF35 therapy.** Since melanoma tumors contain antigen-presenting cells which constitutively express CD40 on their surface, we hypothesized that the activation of these APCs with the CD40 agonist ISF35 might induce tumor-specific CD8 T cell immunity and effective treatment of local and disseminated tumors. Intratumoral injection of

ISF35 into s.c. B16.F10 melanomas significantly suppressed tumor growth and prolonged the survival of mice compared to either control-recombinant adenovirus or saline control-treated mice ($P < 0.05$, unpaired $t$ test, Fig. 1a–c). ISF35 also inhibited tumor growth and prolonged mouse survival significantly better than the intratumoral-injected agonist CD40 mAb ($P = 0.028$, unpaired $t$ test; Fig. 1c). To ensure that antitumor activity of ISF35 was not limited to B16 melanoma, we also used ISF35 to treat established BP melanomas derived from the $Tyr::CreER$; $Braf^{CA}$; $Pten^{lox/lox}$ mouse[13] and we found antitumor activity against these tumors as well (Supplementary Fig. 1).

Analysis of tumor-infiltrating immune cells showed the expansion of the absolute number of myeloid cell types with an activated immunophenotype (Fig. 1d–f). ISF35-treated tumors also contained higher levels of CD8 T cell-related cytokines (IFN-γ, TNF-α, IL-6, IL-13, and GM-CSF), T cell activation markers (4-1BB), and chemokines (CCL4 and CCL5); however, IL-2 and IL-10 cytokines level did not change compared to empty rAd virus or PBS-treated tumors (Fig. 1g). We did not find any upregulation of CD8$^+$ T cell cytokines/chemokines in tumors injected with "empty" rAd vectors, compared to PBS-treated tumors (Fig. 1g), suggesting that rAd itself does not induce a strong CD8$^+$ T cell response in this setting.

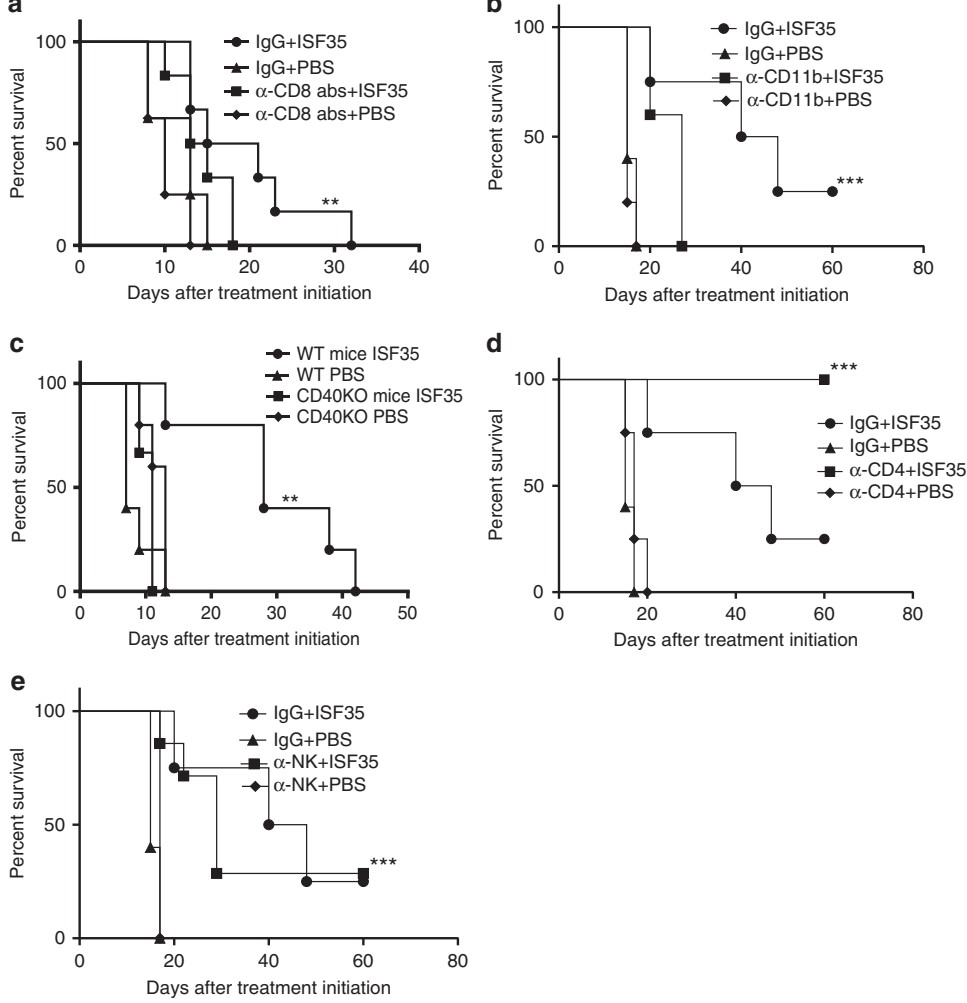

**Fig. 3** Mechanism of action of intratumoral ISF35-mediated antitumor activity. Mice bearing 8-day s.c. B16.F10 tumors were treated every 4 days with ISF35 or PBS and depleting antibodies as indicated. The graphs show the survival of **a** CD8 T cell and **b** CD11b cell-depleted **c** CD40 KO mice **d** CD4 cell and **e** NK cell depleted and after ISF35 treatment. Survival analysis was performed with the log-rank test ($n = 8$). **$P < 0.005$, ***$P < 0.0005$. Error bars are SEM

**Mechanism of tumor suppression**. We investigated the tumor specificity of the T cell response by analysis of tumor antigen-specific T cell activation toward the endogenous retroviral gene product p15E that is naturally expressed by B16 cells[14]. Nine days following i.t. delivery of ISF35, we found significantly greater p15E-specific IFN-γ-producing CD8 T cells compared to control virus-treated mice (Fig. 2a). These CD8 T cells exhibited an effector phenotype, characterized by a low expression of CD62L (L-selectin) (Fig. 2b). We also found an increased expression of T-bet (Fig. 2c) master transcription factors for the generation of effector CD8 T cells, including IFN-γ production[15]. In contrast, we found reduced numbers of suppressive CD4+FoxP3+ T regulatory cells (Treg) (Fig. 2d). p15E-specific CD8+ T cells were also increased in the spleen (Fig. 2e), but splenic Tregs (CD4+FoxP3+) did not change (Fig. 2f).

Tumor growth inhibition following ISF35 treatment was abrogated in CD8 T cell-depleted mice (Fig. 3a), myeloid cell-depleted mice (Fig. 3b), and CD40 knockout mice (Fig. 3c), indicating a requirement for CD8+ T cells, myeloid cells, and CD40 signaling in host cells. Interestingly, the efficacy of ISF35 treatment was increased in CD4 T cell-depleted mice (Fig. 3d), possibly due to the removal of CD4+ regulatory T cells (Tregs). We did not observe any contribution of the NK cell in ISF35-induced antitumor effect (Fig. 3e).

**ISF35 combination therapy with checkpoint inhibitors**. Although intratumoral ISF35 induced a robust expansion of CD8 T cells and suppressed melanoma growth, this monotherapy was not curative. We observed that the T cell checkpoint receptor, PD-1, was upregulated in more than 70% of the tumor-associated CD8 T cells (Fig. 4a), and PD-1 ligand (PD-L1) was expressed on the majority of tumor-associated myeloid cells following ISF35 therapy (Fig. 4b), suggesting that PD-1 engagement could limit antitumor activity. Indeed, combining i.t. ISF35 with systemic anti-PD-L1 antibody significantly increased mouse survival, compared to either monotherapy ($P < 0.05$, log-rank test; Fig. 4c). Although ISF35 did not directly induce a greater expression of CTLA-4 on CD8 T cells (Fig. 4d), ISF35 plus anti-CTLA-4 combination therapy induced a modestly increased survival compared to ISF35 monotherapy and a significantly greater survival compared to anti-CTLA-4 monotherapy (Supplementary Fig. 2a). Furthermore, ISF35 plus anti-PD-1 significantly increased the fraction of CTLA-4-expressing CD8 T cells (Fig. 4d). The treatment of mice with the triple combination of ISF35, anti-CTLA-4, and anti-PD-1 antibody significantly prolonged mice survival, with 40% of the mice cured ($P = 0.0002$, log-rank test; Fig. 5a). Cured mice also resisted a tumor rechallenge, indicating protective antitumor immunity after combination therapy (Fig. 5b). Vitiligo, a marker for T cell immunity

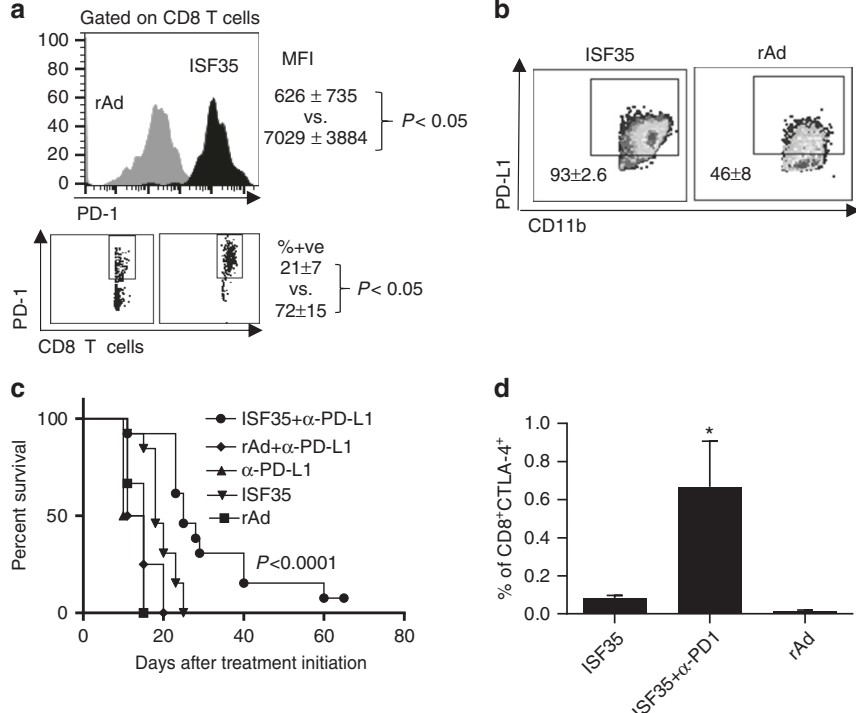

**Fig. 4** Antitumor effect of intratumoral ISF35 and anti-PD-1 antibody combination therapy. Mice bearing 8-day s.c. B16.F10 tumors were treated as indicated, and tumor-infiltrating leukocytes were stained after 5 days of treatment and analyzed by flow cytometry for the expression of PD-1 and PD-L1. **a** Mean fluorescence intensity (MFI) and percentage of CD45$^+$CD8$^+$PD-1$^+$cells. **b** Percentage of CD45$^+$CD11b$^+$PD-L1$^+$ cells. **c** Mice bearing 8-day s.c. B16.F10 tumors were treated with i.t. ISF35 or rAd and anti-PD-L1. The graph depicts tumor growth at different time points. **d** Percentage of CD8$^+$CTLA-4$^+$ cells after indicated treatment. Data are analyzed by unpaired two-tailed $t$ test or one-way ANOVA. *$P < 0.05$, **$P < 0.005$ ($n = 4$–8 mice). Error bars are SEM. Survival analysis was performed with the log-rank test

against melanocyte differentiation antigens[16], was also observed following triplet combination treatment (Fig. 5c). Finally, tumor-infiltrating T cells following the triple combination contained a decreased percentage of Tregs (5d), increased percentage of CD8 T cells (Fig. 5e), and a higher ratio of CD8 T cells to Tregs (Fig. 5f).

**Systemic tumor regression after ISF35 and checkpoint blockade.** The ultimate goal of local immunotherapies such as intratumoral ISF35 is to induce systemic immunity to affect systemic tumor regression, including at distant uninjected sites (abscopal effect). We tested this principle by treating mice bearing 2 tumors on the right and left flank with ISF35 only in the right flank tumor, followed by systemic anti-CTLA-4 and anti-PD-1 mAbs. We implanted a distant tumor 3 days later than a primary (drug-injected) tumor, since we were mimicking micrometastases (Fig. 6a). Triple-combination therapy suppressed both the injected and the distant uninjected tumor, and prolonged mouse survival compared to ISF35 monotherapy or dual-checkpoint blockade (Fig. 6b, c). The triple combination also increased tumor-specific CD8 T cells systemically compared to ISF35 monotherapy or anti-CTLA-4 plus anti-PD-1 double therapy (Fig. 6d), supporting the role of tumor-specific CD8 T cells (Fig. 3a) in both local and distant tumor suppression. These results indicate that i.t. ISF35 in combination with PD-1 and CTLA-4 blockade suppresses not only the directly injected tumor, but also a distant, uninjected tumor. Double combination of i.t. ISF35 with a systemic anti-PD-1 antibody without anti-CTLA-4 antibody, significantly inhibited tumor growth of both injected and uninjected tumor compared to either monotherapy ($P < 0.05$,

unpaired $t$ test; Supplementary Fig. 2b); however, in contrast to triple combination, dual combination was unable to cure the mice from tumor (Fig. 6c and Supplementary Fig. 2b).

Next, we evaluated whether ISF35 might leak out of the injected tumor and enter into the uninjected, distant tumor through the circulation, thus causing CD40L expression and downstream effects in both tumors. Upon i.t. injection of ISF35, the expression of CD40L in both immune and tumor cells was found only in treated tumor but not in a distant untreated tumor (Fig. 6e), suggesting that ISF35 induced a local CD40L expression, causing a systemic T cells response against both injected and uninjected, distant tumors.

**Combination therapy eradicates melanoma in the skin and the brain.** The brain represents a unique immunological environment that includes a specialized vasculature and a cytokine environment that protects the brain from excessive inflammation and T cell infiltration[17, 18]. Cancer metastatic to the brain is generally difficult to treat, with high rates of morbidity and mortality. We evaluated whether T cells, induced by local therapy of a s.c. melanoma with i.t. ISF35, could clear melanoma tumors in the brain. B16 tumors were implanted with both s.c. and directly into the brain, followed by i.t. injection of ISF35 in the s.c. tumor and i.p. administration of anti-PD-1 and anti-CTLA-4 (Fig. 7a). The triple-combination therapy was significantly more effective than ISF35 monotherapy or doublet checkpoint inhibitor therapy in suppressing both the s.c.-injected tumor and the uninjected brain melanoma (Fig. 7b–d), prolonging the mouse survival and resulting in a 45% complete cure rate.

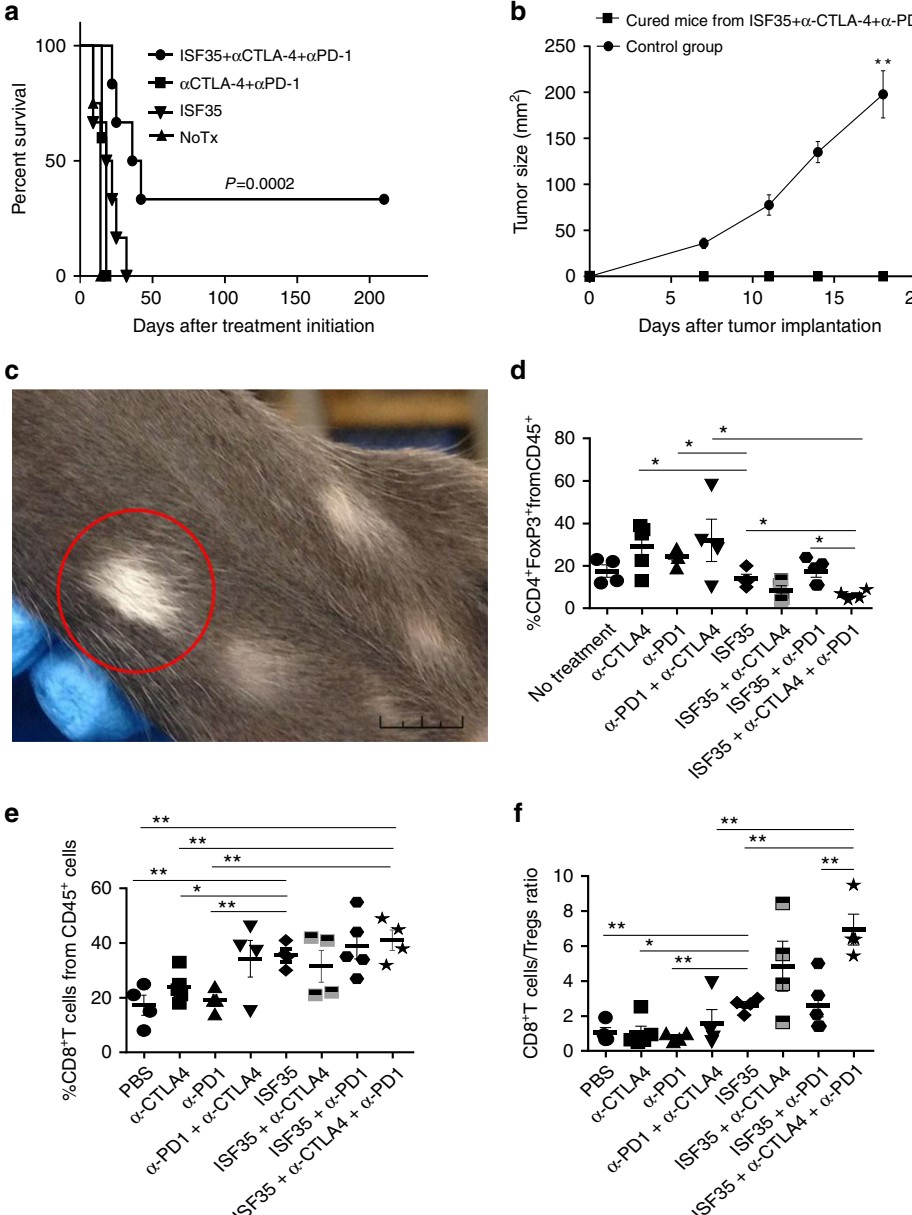

**Fig. 5** Synergistic effect of intratumoral ISF35 with anti-CTLA-4 and anti-PD-1 antibody blockade. Mice bearing 8-day s.c. B16.F10 tumors were treated as indicated. **a** Mouse survival. **b** Tumor growth after rechallenge of cured mice. **c** Vitiligo in a cured mouse (1 cm; represents 3 mice). Mice bearing 8-day s.c. B16.F10 tumors were treated as indicated, and 9 days later, tumors were analyzed for the presence of **d** Tregs (CD4+FoxP3+) (**e**) CD8+ T cells and **f** CD8 to Treg ratio. Data are analyzed by unpaired two-tailed $t$ test. *$P < 0.05$, **$P < 0.005$ ($n = 4$–8 mice). Error bars are SEM

## Discussion

At present, the success of immunotherapy for melanoma appears to depend on enhancing melanoma-specific CD8+ T cell immunity since CD8+ T cells are strongly associated with direct tumor killing and a melanoma patient's survival. Thus, therapeutic modalities that promote CD8+ T cell responses are a key goal in cancer immunotherapy drug development.

CD4+ T cells enhance CD8+ T cell priming by licensing dendritic cells (DCs) via CD40–CD154 interactions. In addition, CD40–CD154 interactions prevent the CD8+ T cell response from diminishing prematurely. Therefore, CD40 activation through CD154 is a requisite step for inducing effective antigen-specific CD8 T cell immunity against pathogens and tumor[19–22]. In this regard, agonistic CD40 antibodies appear as a promising strategy for cancer immunotherapy. However, clinical trials and our present study using such antibodies show a limited efficacy

and sometimes toxicity, highlighting the importance of alternative approaches to therapeutic CD40 ligation.

ISF35 has been tested in multiple clinical studies and was found to be safe and effective[12]. Specifically, preexisting antibodies against Ad 5 did not abrogate Ad-ISF35 antitumor activity, and intratumoral administration of Ad-ISF35 in patients did not cause any toxicities or vector accumulation[11, 12]. As melanoma is an immunogenic tumor containing CD40-expressing myeloid cells including DCs, we tested the efficacy of ISF35 against melanoma and observed a strong efficacy against highly aggressive and established B16 melanoma. The efficacy of ISF35 greatly increased when combined with CTLA-4 and PD-1 blockades, resulting in increased overall survival and cure. In addition, this combination therapy induced systemic, tumor-specific CD8+ T cell immunity, resulting in complete and durable abscopal regression of uninjected contralateral and brain tumors. Given

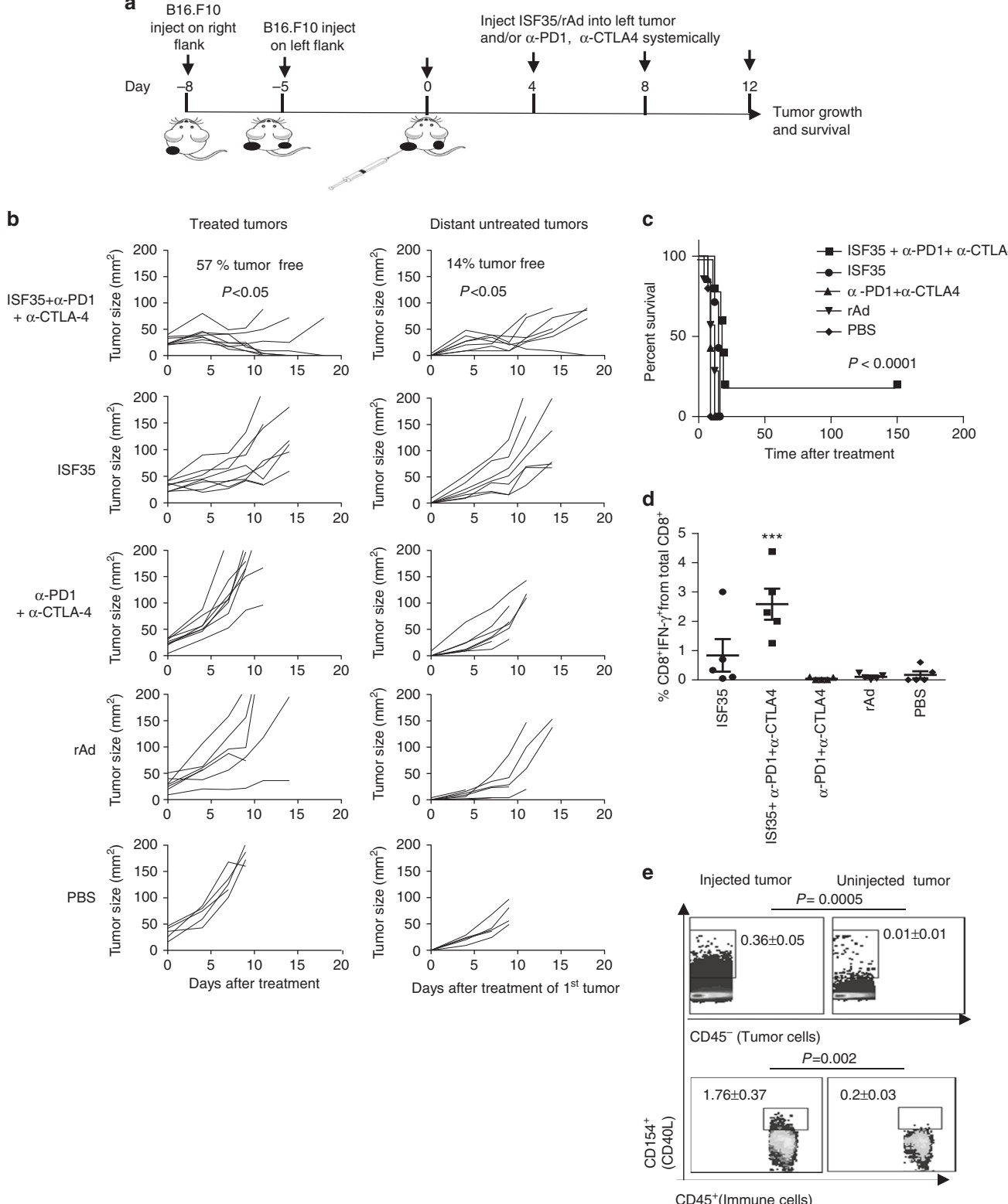

**Fig. 6** Abscopal effect after intratumoral ISF35 with anti-CTLA-4 and anti-PD-1 antibody blockade. **a** Treatment strategy. **b** Growth of injected and distant, uninjected B16.F10 tumors ($n = 5$–7). **c** Mouse survival ($n = 5$–7). **d** Tumor antigen (p15E)-specific IFN-γ producing CD8 T cells in circulation ($n = 5$). **e** Expression of CD40L on immune cells and tumor cells in treated and untreated tumor after 24 h of ISF35 treatment ($n = 3$). Data are representative of at least two independent experiments and analyzed by unpaired two-tailed $t$ test or one-way ANOVA. *$P < 0.05$, **$P < 0.005$ and ***$P < 0.0005$. Error bars are SEM. Survival analysis was performed with the log-rank test

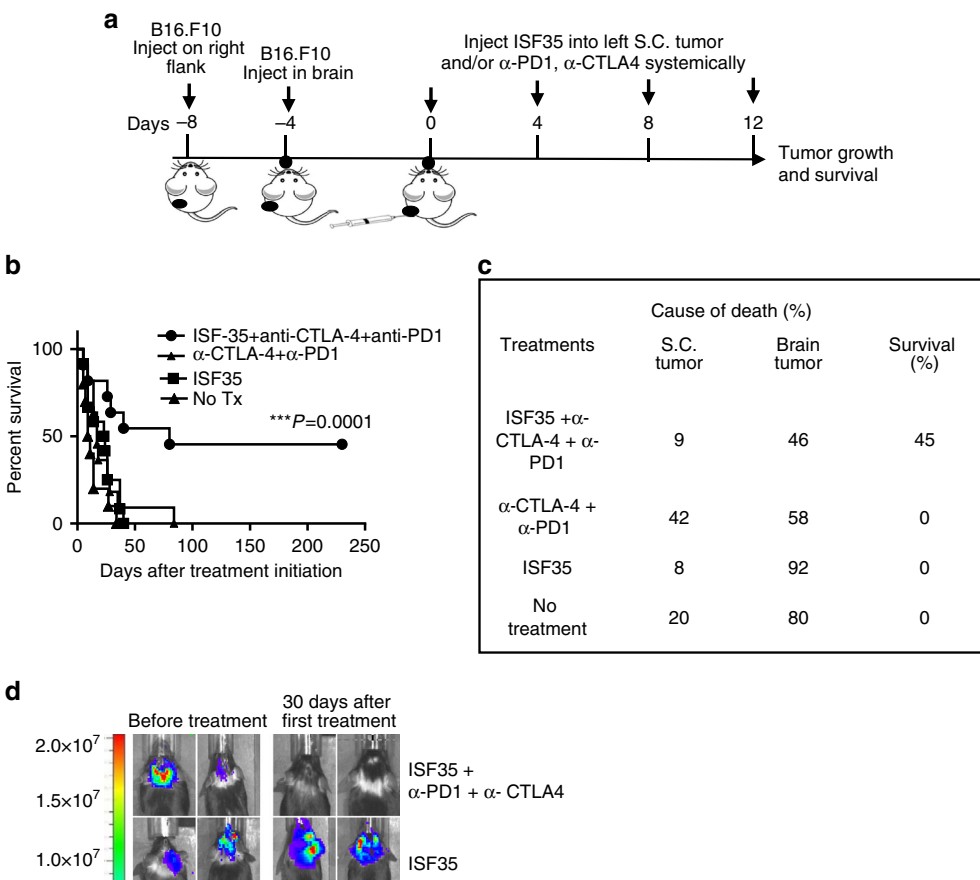

**Fig. 7** Synergistic effect of intratumoral ISF35 with anti-CTLA-4 and anti-PD-1 antibody therapy against melanoma in the brain. **a** Treatment strategy. **b** Mouse survival. **c** Cause of death (s.c. or brain tumor). **d** Representative luminescence images from treated animals. Data are from two pooled independent experiments (*n* = 7 in each experiment). Survival analysis was performed with the log-rank test. ***P < 0.0005

these results, patients with metastatic melanoma may benefit from intratumoral ISF35 therapy in combination with systemic anti-PD-1 and/or anti-CTLA-4 therapy. Indeed, intratumoral administration of FDA-approved T-VEC (GM-CSF-expressing Herpes simplex virus-1) showed a limited efficiency against more advanced melanoma[23–25] unless combined with checkpoint blockade[26].

A large fraction of patients with cancer, including melanoma, do not experience tumor regression after single-checkpoint or dual-checkpoint blockade with anti-PD-1 and anti-CTLA-4 mAbs, likely due in part to insufficient spontaneous antitumor T cell immunity. Indeed, mice bearing established B16 melanoma showed no induction of melanoma-specific CD8 T cells after dual-checkpoint blockade, and did not experience tumor regression. The addition of i.t. ISF35-induced melanoma-specific CD8 T cells, and caused tumor regression. Thus, i.t. ISF35 appears to act as an inducer of tumor-specific CD8 T cell immunity, which is then amplified and functionally sustained by dual-checkpoint blockade.

Upregulation of PD-1 on CD8 T cells after ISF35 treatment, and upregulation of CTLA-4 on CD8 T cells after ISF35+anti-PD-1 treatment, likely explain the better efficacy of triple combination than mono or dual therapy. A higher CD8/Treg ratio in ISF35+anti-CTLA-4+anti-PD-1-treated mice tumors support the notion that a higher CD8/Treg ratio is associated with better treatment efficacy and a favorable clinical outcome in preclinical

models and cancer patients[27, 28]. However, the mechanism of intratumoral Treg depletion is not clear. It is possible that the expansion of CD8 T cells through ISF35 within tumor may be responsible for the increased CD8/Treg ratio. However, since we also saw more CD8+ T cells in the spleen, yet no change in CD8/Treg ratio, other mechanisms likely contribute, including possible changes in Treg trafficking, or intratumoral Treg proliferation and/or apoptosis.

A weaker efficacy of "empty" control adenovirus and no effect of ISF35 treatment in CD40 KO mice suggests that CD40 ligation is essential for the activity of ISF35. However, it is possible that the inflammation induced by adenoviral infection potentiates the activity of CD40 ligation, supported by the finding that intratumoral anti-CD40 mAb was not as effective as CD40L delivered by adenoviral infection. Many tumor cells express CD40, including almost 100% of B-cell malignancies and up to 70% of solid tumors, including melanoma[29–31]. Moreover, signaling mediated by CD40 stimulation can induce cell death in cancer cells, making this pathway an attractive target for cancer immunotherapy. However, ISF35 did not work in the absence of CD40 in host cells, suggesting that any direct killing of B16 melanoma by CD40 signaling is insufficient to induce therapeutic antitumor immunity, and instead highlighting the importance of CD40 ligation on host cells.

Melanoma brain metastases (MBM) are a rapidly growing clinical problem with up to 60% of melanoma patients developing

MBM over the course of their disease. Locoregional treatment with surgery, radiotherapy, radiosurgery, and newer drug classes such as checkpoint inhibitors and targeted agents against BRAF-mutation melanoma have shown limited effectiveness against MBM[32]. MBM patients continue to have poor clinical outcomes and life expectancies of only 3–7 months[33]. In this regard, ISF35 in combination with anti-CTLA-4 and anti-PD-1 checkpoint antibodies shows a promising therapeutic efficacy against melanoma in the brain. It has been reported that checkpoint blockade antibodies do not cross the blood–brain barrier, although activated T cells in the periphery can migrate to the brain[34] and mediate intracerebral tumor regression.

In summary, intratumoral administration of ISF35 generates antitumor immune responses mediated through host cell CD40 signaling and CD8 T cells. The addition of ISF35 overcomes the primary resistance to the checkpoint blockade antibodies, anti-PD-1 and anti-CTLA-4, and generates potent systemic antitumor T cell immunity that eradicates the injected tumor as well as distant, uninjected tumors, including the brain. This preclinical study paves the way for clinical trials of ISF35 and checkpoint blockade for patients with metastatic cancer, including the brain.

## Methods

**Mice and cell lines**. All animal experiments were performed in accordance with National Institutes of Health Guidelines and approved by the MD Anderson Cancer Center Institutional Animal Care and Use Committee. C57BL/6 and CD40 KO mice were purchased from Jackson Laboratory. All female mice were used at 6–12 wk of age. Mycoplasma-free B16.F10 was obtained from ATCC and the braf$^{V600E}$×pten$^{-/-}$ (BP)[13] melanoma cell line was kindly provided by Dr Patrick Hwu, MD Anderson Cancer Center, Houston, TX. Both cell lines were maintained in RPMI 1640 supplemented with 10% of heat-inactivated FBS, L-glutamine, sodium pyruvate, nonessential amino acids, and penicillin–streptomycin (all from Invitrogen/Life Technologies).

**Tumor induction and monitoring**. C57BL/6 or KO mice were s.c. inoculated with $5 × 10^5$ B16.F10 melanoma cells on day–8 in the right flank of treatment. The same amount of B16.F10 cells was also inoculated into the left flank on day–5 for contralateral tumor experiments. For the brain metastases model, $5 × 10^5$ B16.F10-luciferase melanoma cells were inoculated on the right flank on day–8. On day–4, $5 × 10^3$ B16.F10-luciferase cells were injected and centered at 2 mm posterior to the coronal suture and 2 mm lateral to the sagittal suture at a depth of 3 mm. Tumor size is expressed as the product of perpendicular diameters of tumors measured with calipers. The mice were killed when tumor size reached ≥ 200 mm$^2$ in diameter. Brain tumors were monitored by in vivo imaging (IVIS spectrum), with tumor size expressed by the luciferin signal analyzed using Living Image 2.6 (Caliper LifeSciences). The mice were additionally monitored by their behavior and killed by an excessive hunched posture or the impairment of motoric capability.

**Treatment with ISF35 and checkpoint blockade**. ISF35 and control Ad5 were provided by Memgen, LLC (Houston, TX). Mouse Abs against CTLA-4 (9H10), PD-1 (RMP1-14), PD-L1 (10F.9G2), CD4 (GK1.5), CD8 (2.43), NK (NK1.1), CD11b (M1/70), and agonist CD40 (FGK4.5) were purchased from Bio X Cell. In every experiment, only the right flank tumor was treated intratumorally (i.t.) twice a week with ISF35 or control virus ($1 × 10^{10}$ viral particles/injection) or 50 µg of agonist CD40 antibody. In total, four treatments were given per experiment. α-PD-1, α-PD-L1, and/or α-CTLA-4 were injected every 3 days i.p. (200 µg), which were given for 4 wk.

**Flow-cytometric analysis**. Leukocytes were isolated from mechanically disrupted tumors by a lymphocyte separation medium (Corning cellgro). RBC lysis was performed on blood. Intracellular IFN-γ APC (clone XMG1.2; 1:400 dilution) staining was performed using Cytofix/Cytoperm kit (BD Biosciences). Transcription factor staining for Foxp3 PE (clone MF23; 1:400 dilution) and T-bet PerCPCy5.5 (clone 04-46; 1:400 dilution) was performed using Foxp3-staining buffer set (Affymetrix eBioscience). Briefly, the cells were first stained for surface antigens CD4 or CD8 followed by washing, fixation, and permeabilization and then stained with Abs against IFN-γ, FoxP3, or T-bet. Cells were surface stained with Abs against CD45 APC/pacific blue (clone 30-F11; 1:200 dilution), CD8 PE/PerCPCy5.5/FITC (clone 53-6.7; 1:200 dilution), CD62L FITC (clone MEL-14; 1:100 dilution), CD4 PerCP (clone GK1.5; 1:200 dilution), CD11b FITC (clone M1/70; 1:100 dilution), CD11c PE-Cy7 (clone N418; 1:400 dilution), CD86 pacific blue (clone GL-1;1:100 dilution), CD40 PE (clone 3/23; 1:200 dilution), CD40L PE-Cy7(MR-1; 1:400 dilution), PD-1 PE-cy7 (clone RMP1-30; 1:400 dilution), PD-L1 brilliant violet 421

(clone 10F. 9G2;1:100 dilution), and CTLA-4 brilliant violet 421 (clone uc10-4B9; 1:100 dilution) from BioLegend. Data were acquired on a Canto II flow cytometer (BD Biosciences) and analyzed using FlowJo (Tree Star).

**Cytokine/chemokine assay**. After 9 days following ISF35 treatment, tumors were mechanically disrupted and centrifuged, and the supernatant was collected. An aliquot of 25 µl of a tumor supernatant from each sample was used to perform the assay. Cytokines/chemokines were measured using a Milliplex mouse cytokine/chemokine panel (Millipore) according to the manufacturer's instructions. A fluorescence signal was measured on a Luminex 100/200 system, and data were analyzed using Excel software.

**Statistical analysis**. All results are expressed as the means ± SEM. For therapeutic experiments, five to ten mice were assigned per treatment group. This sample size gave 80% power to detect a 50% reduction in tumor size with 95% confidence. Statistical analysis was performed with GraphPad Prism 6 software. Data were analyzed using unpaired Student's t test, and ANOVA and its differences were considered to be significant at $P < 0.05$. Survival experiments used log-rank Mantel–Cox test for analysis. All experiments were performed at least twice with comparable results.

**Data availability**. The data supporting the findings of this study are available within the article and its Supplementary Information files and from the corresponding author upon request.

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

## Acknowledgements

This work was supported by Memgen, LLC (to W.W.O.) and the University of Texas MD Anderson Cancer Center SPORE in Melanoma P50CA093459 (to M.Si.).

## Author contributions

Conception and design: M.Si. and W.W.O. Development of methodology: M.Si., C.V., Z. D., Z.X., M.Sh., H.K., F.F., Y.H., A.R.J., L.M.E.J., U.B. and R.L.B. Acquisition of data: M. Si. and C.V. Analysis and interpretation of data: M.Si. Critical comments on manuscript: D.J.T., A.D., M.A.C. and P.H. Writing, review, and/or revision the manuscript: M.Si., W.W.O. and M.J.C. Study supervision: W.W.O.

## Additional information

**Competing interests:** M.J.C. is an employee of Memgen and inventor on patents and patent applications concerning the composition of matter and use of ISF35 for cancer therapy. W.W.O., M.Si., P.H. and M.J.C. are the authors and inventors on U.S. patent application "Methods and therapeutic combinations for treating tumors" No. 15/500,618 filed on 31 July 2015, concerning the use of ISF35 for cancer therapy. The remaining authors declare no competing financial interests.

