## [Peer Review File · Nature Communications]

Reviewer #2 (Remarks to the Author):

The manuscript submitted to Nature Communications by Singh et al, entitled "Intratumoral CD40 activation and checkpoint blockade induces systemic CD8+ T cell immunity that eradicates distant tumors, including in the brain" describes a series of experiments in mice with B16F10 melanoma in which the authors utilize a novel adenovirus construct that encodes a trimeric CD40 ligand molecule that binds to CD40 on antigen-presenting cells and possibly also tumors. They demonstrate that addition of that construct to CTLA-4 and PD-1 antibodies promote cures of B16F10 and have a significant impact on brain metastases derived from that tumor, a very important and intriguing result with clinical translational implications. Surprisingly, many of the control experiments in figures 1 through 5 seem incomplete, since they lack the inclusion of the empty adenovirus, and it is shown that the empty adenoviral construct does have some anti-tumor activity when injected intra-tumorally, which necessitates the essential controls noted above. Overall, this is a useful work that was well performed and might be acceptable for inclusion in Nature Communications with important changes including the inclusion of improved controls in the experiments of figure 1, 2, 3 and 4 as detailed below. Also, the authors need to address the very practical issue of using an adenoviral vector in humans with immune reactivity to those vectors. In detail:

Why were the right and left flank injections done on different days?

The data in figure 1G should show the equivalent cytokine levels after adenovirus control injection, not just PBS

The data in figures 2D, 2E and 2F should show equivalent T cell numbers and ratios of T cell subsets to T regs after adenovirus control injection, not just PBS

The data in figure 3C and 3D should also include the empty adenoviral controls

The survival data in figure 4A should also include the empty adenoviral controls

The data in figures 5B, C and D should also include the empty adenoviral controls

Reviewer #3 (Remarks to the Author):

The authors look at the combination of an intratumoral Adenovirus gene therapy vector expressing CD40L (ISF35) in combination with anti-PD1 and anti-CTLA4 in the treatment of metastatic melanoma in mouse tumor models, with the triple combination demonstrating complete responses against both treated and untreated tumors. The approach is of limited novelty, as many groups are looking to combine different immunotherapies that have previously failed in the clinic with checkpoint inhibitors. In this case, although CRs in the one model tested requires a triple combination of biological agents, the logical sequence of addition of the different components, the reasonable responses, especially in brain 'metastases' and the translational potential make this of interest.

There are several comments on the data provided;

The mechanistic data only examines CD8+ T-cells. Although this is the expected effect of the CD40L it would be of interest to know the importance of other cell types, such as CD4+, NK and myeloid cells in the therapeutic effects through depletion or transgenic knock out experiments.

Does the effect induce a potent anti-Ad CD8+ T-cell response as well?

It would also be of interest to examine a second tumor model.

We wish to thank the reviewers for their generous and insightful comments which have helped us to further strengthen the manuscript.

Reviewer #2:

The authors need to address the very practical issue of using an adenoviral vector in humans with immune reactivity to those vectors.

We agree that this is an important point for clinical application, and it was previously addressed for ISF35. Specifically, pre-existing antibodies against Ad-5 did not abrogate Ad-ISF35 anti-tumor activity and intratumoral administration of Ad-ISF35 did not cause any toxicities or vector accumulation (Melo-Cardenas J etal 2012 and Castro JE 2012). We have added a discussion of this important point to the manuscript, Page# 7, paragraph 3.

Why were the right and left flank injections done on different days

We implanted distant tumor 3 days later than primary (drug-injected) tumor, since we were mimicking micrometastases. We hypothesized that anti-tumor immunity that induce in primary tumor in response to ISF35 treatment could take some time to have systemic effect and shrink distant tumor. We have included this consideration in the results section Page # 6, paragraph 1.

The data in figure 1G should show the equivalent cytokine levels after adenovirus control injection, not just PBS

We have performed this new experiment and show the data in figure 1G with cytokine levels after empty adenovirus (rAd) injection.

The data in figures 2D, 2E and 2F should show equivalent T cell numbers and ratios of T cell subsets to T regs after adenovirus control injection, not just PBS

We have performed these new experiments and replaced old figures 2D, 2E and 2F with new figures 2D, 2E and 2F including empty adenovirus (rAd).

The data in figure 3C and 3D should also include the empty adenoviral controls.

We have performed these new experiments to include empty adenovirus (rAd) in figures 4C and 4D (in this revision, these are 4C and 4D instead of 3C and 3D).

The survival data in figure 4A should also include the empty adenoviral controls

The data in figures 5B, C and D should also include the empty adenoviral controls
Since both experiments in figures 4A (single tumor) and 5C (double tumors) received exactly the same treatment, and we already had data on the empty adenovirus control (rAd) for Fig 5, we have now updated Fig. 5 with those data, shown as Fig. 6B, 6C and

6D in the revised version of the paper.

Reviewer #3:

The mechanistic data only examines CD8+ T-cells. Although this is the expected effect of the CD40L it would be of interest to know the importance of other cell types, such as CD4+, NK and myeloid cells in the therapeutic effects through depletion or transgenic knock out experiments.

We have performed these new experiments and included these data in Fig. 3B, C and D.

Does the effect induce a potent anti-Ad CD8+ T-cell response as well?

It has been reported that Ad vectors can generate CD8+T cell responses directly. We did not find any upregulation of CD8+ T cell cytokines/chemokines in tumors injected with “empty” rAd vectors, compared to PBS-treated tumors (Fig. 1G), suggesting that rAd itself does not induce a strong CD8+ T cell response in this setting. We have included this consideration in the results section Page # 4, paragraph 3.

It would also be of interest to examine a second tumor model.

We performed new experiment using the BP melanoma model, derived from BRAF^{V600E}xPTEN^{-/-} melanoma prone-mice added the results as Fig. S1.

Reviewer #2 (Remarks to the Author):

The authors have done a nice job of responding to the reviewers' requirements and comments, and have included virtually all of the requested additional data, and provided explanations for all of the queries.

Reviewer #3 (Remarks to the Author):

The authors have adequately addressed my previous concerns